# Development of a Human B7-H3-Specific Antibody with Activity against Colorectal Cancer Cells through a Synthetic Nanobody Library

**DOI:** 10.3390/bioengineering11040381

**Published:** 2024-04-15

**Authors:** Jingxian Li, Bingjie Zhou, Shiting Wang, Jiayi Ouyang, Xinyi Jiang, Chenglin Wang, Teng Zhou, Ke-wei Zheng, Junqing Wang, Jiaqi Wang

**Affiliations:** 1School of Pharmaceutical Sciences (Shenzhen), Shenzhen Campus of Sun Yat-sen University, Shenzhen 518107, China; lijx339@mail2.sysu.edu.cn (J.L.); zhoubj9@mail2.sysu.edu.cn (B.Z.); wangsht35@mail2.sysu.edu.cn (S.W.); ouyjy6@mail2.sysu.edu.cn (J.O.); jiangxy227@mail2.sysu.edu.cn (X.J.); wangjunqing@mail.sysu.edu.cn (J.W.); 2Shenzhen Qiyu Biotechnology Co., Ltd., Shenzhen 518107, China; wangchenglin0510@outlook.com; 3School of Cyberspace Security, Hainan University, Haikou 570228, China; teng.zhou@hainanu.edu.cn; 4School of Biomedical Sciences, Hunan University, Changsha 410082, China; zhengkewei@hnu.edu.cn

**Keywords:** synthetic library, nanobody, phage display, B7-H3, ADCC

## Abstract

Nanobodies have emerged as promising tools in biomedicine due to their single-chain structure and inherent stability. They generally have convex paratopes, which potentially prefer different epitope sites in an antigen compared to traditional antibodies. In this study, a synthetic phage display nanobody library was constructed and used to identify nanobodies targeting a tumor-associated antigen, the human B7-H3 protein. Combining next-generation sequencing and single-clone validation, two nanobodies were identified to specifically bind B7-H3 with medium nanomolar affinities. Further characterization revealed that these two clones targeted a different epitope compared to known B7-H3-specific antibodies, which have been explored in clinical trials. Furthermore, one of the clones, dubbed as A6, exhibited potent antibody-dependent cell-mediated cytotoxicity (ADCC) against a colorectal cancer cell line with an EC50 of 0.67 nM, upon conversion to an Fc-enhanced IgG format. These findings underscore a cost-effective strategy that bypasses the lengthy immunization process, offering potential rapid access to nanobodies targeting unexplored antigenic sites.

## 1. Introduction

Antibodies have made a revolutionary impact in the realms of science and medicine because of their remarkable specificity and biochemical adaptability, making them applicable across a wide spectrum of biomedical investigations. The conventional antibody structure comprises two heavy chains and two light chains, each housing a variable domain denoted as VH and VL, respectively. A notable deviation from this typical architecture is observed in camelids (llamas, camels, alpacas, and their kins), which possess an antibody repertoire exclusively composed of heavy chains [1]. In this unique system, antibodies bind to their target antigens using a single variable domain of heavy-chain-only antibody (VHH, also known as a nanobody). A nanobody has a smaller size (∼10–15 kDa) compared to the traditional fragment of antigen binding (Fab) (∼50 kDa). In addition, it does not rely on heavy–light chain pairing, which renders it highly suitable for constructing polyspecific antibodies from a protein engineering perspective. Unlike the Fab of conventional antibodies, a nanobody can be efficiently expressed in bacteria. The versatility of nanobodies extends their utility to various applications in protein structural biology and cell biology, positioning them as potential diagnostic and therapeutic agents [2].

Despite the increasing significance of nanobodies in biomedical research, existing methods for producing monoclonal nanobody sequences involve the immunization of camelids, which remains tedious, costly, and frequently unavailable. Additionally, antibodies obtained from immunization often face limitations in binding immunodominant epitopes. The identification of functional clones, such as conformationally selective nanobodies, remains a challenging task [3,4]. In response to these challenges, synthetic libraries have been designed for the in vitro selection against diverse antigens [4,5,6,7]. In this study, we also devised a synthetic nanobody library in a phage display format for the in vitro selection of antigen-specific nanobodies. Unlike most of the other synthetic nanobody libraries, selected complementarity determining region (CDR) positions in our library were randomized using degenerate codon. This design performs complete randomization of these sites, which may allow for the selection of CDR sequences not captured by a pre-biased library with a planned diversity introduced by a combination of trinucleotide phosphoramides [3,4,5,6,8,9].

To validate the efficacy of our proposed libraries, we executed antibody discovery against an anti-tumor drug target: CD276 (also denoted as B7-H3). It is a member of the B7 ligand family, emerging as a promising target for antibody-based immunotherapy [10,11]. The prevailing variant of human B7-H3 features four immunoglobulin-like domains (Ig) in a tandemly repeated format such as IgV-IgC-IgV-IgC, while human B7-H3 exists in two isoforms (2IgB7-H3 and 4IgB7-H3); its mouse counterpart has a singular 2Ig isoform [12]. B7-H3 is prominently expressed in both differentiated malignant cells and cancer-initiating cells, exhibiting limited heterogeneity but a high frequency across various cancer types [11]. Importantly, its expression in normal tissues is low. In nonmalignant tissues, B7-H3 primarily assumes an inhibitory role in adaptive immunity by suppressing T-cell activation and proliferation [13,14]. In malignant tissues, B7-H3 hampers tumor antigen-specific immune responses, fostering a tumor-promoting effect [11]. Beyond its immunologic functions, B7-H3 also facilitates migration and invasion [15], promoting angiogenesis [16,17], enhancing chemoresistance [18,19], influencing tumor cell metabolism [20,21], and triggering endothelial-to-mesenchymal transition [22,23]. Consequently, the presence of B7-H3 in tumors correlates with a poor prognosis [17,24]. Despite experimental evidence indicating that silencing B7-H3 reduces the malignant potential of cancer cells, the development of B7-H3-blocking antibodies has received limited attention, largely due to the unknown B7-H3 receptor. Instead, numerous antibody-based strategies have been devised leveraging distinct effector mechanisms against B7-H3-expressing cancer cells [11].

Antibodies targeting distinct epitopes can show different modes of actions and affect potencies [25,26]. Both the mouse-derived anti-human B7-H3 monoclonal antibodies (MAbs) 8H9 and 376.96, which underwent clinical testing, were identified as binding to the FG loop region at the top of the IgV domain [27,28]. The single-chain structure of nanobodies contributes to the convex shape of the paratopes. In comparison, the paratopes of conventional antibodies are generally wider and form a flat surface, a groove or cavity due to the heavy and light chain pairing. Thus, the footprint of the paratope of a nanobody on its cognate antigen might be different from that of conventional antibodies. In this study, we took advantage of the synthetic nanobody library, aiming to discover B7-H3-specific nanobodies which might bind to different epitopes compared to known MAbs. The synthetic phage display nanobody library was first evaluated by next-generation sequencing (NGS) to ensure diversity. A phage display selection was then conducted to enrich binders for recombinant human B7-H3 proteins. Both single-clone phage enzyme-linked immunosorbent assay (ELISA) and NGS were applied to identify binders. Subsequently, the positive clones were characterized for their affinity, specificity, and potential binding sites. Lastly, one of the nanobodies with a higher binding affinity was converted to a human IgG1 fragment crystallizable (Fc) domain-fusion antibody and tested in an antibody-dependent cellular cytotoxicity (ADCC) reporter assay to demonstrate its functional activity. As outlined in this report, the described library exhibits high productivity and holds the potential to generate specific MAbs targeting distinct epitopes.

## 2. Results

### 2.1. Analysis of the Synthetic Library Sequences

Assembly PCR was used to synthesize DNA sequences, which encoded a stable framework region of a nanobody and three randomized CDR regions (CDR1, CDR2, and CDR3). A significant difference between our design and others’ is the usage of the degenerate codon “NNB” to randomize desired sites, instead of using mixed trinucleotide phosphoramidites (Figure 1A). This design introduces fewer biases from the planned percentages of trinucleotides and also requires much fewer costs for its construction compared to the mixed trinucleotide method. In addition, some sites were partially randomized by mixed nucleotides to allow for up to six possible amino acids. CDR3 loops were designed to have three different lengths, including 7 (short), 11 (medium), and 15 (long) fully randomized consecutive sites. The assembled nanobody genes were ligated into pMES4 vectors, which were then used to transform *E. coli* as described in the methods. The production of the phage display nanobody library was conducted by infecting the *E. coli* library with an M13KO7 helper phage. To evaluate the quality of the library, NGS was conducted to analyze the assembled sequences. The ratios of amino acids were close to the expected ones based on the NNB encoding at each position (Figure 1B,C). The percentages of sequences with short/medium/long CDR3 reached 3:1:1 in the final phage library. As the library assembly through PCR could introduce insertions, deletions, or frame shifts in the final product, the percentage of complete and in-frame sequences in the library was quantified. The findings indicated that the proportion of intact sequences was roughly 40% in the transformed *E. coli* library and approximately 35% in the final phage library (Figure 1D). A 3×Myc tag was intentionally inserted between the C-terminus of the nanobody and pIII protein of the M13 phage. By one-step enrichment using an anti-Myc tag antibody, the percentage of intact nanobody genes was improved to 80% (Figure 1D). To assess the diversity, the occurrences of each unique nanobody sequence within the synthetic library were quantified. The result revealed a pronounced degree of diversity for the displayed repertoire (Figure 1E). Approximately 90% of the sequences exhibited no more than three occurrences in the NGS datasets of both the *E. coli* and phage libraries. The proportion slightly decreased to approximately 80% after the anti-Myc tag enrichment process.

### 2.2. Protein-Based Panning for B7-H3-Specific Nanobody

To illustrate the functionality, the constructed phage display nanobody library was used to pan against recombinant human 4IgB7-H3, which contained two V-type and two C-type Ig domains. The panning process was carried out first in B7-H3 protein-coated Maxisorp plate for two rounds. Bound phages were eluted using glycine buffer at a low pH and amplified for the following rounds of the selection process. To reduce unspecific binders, in the third round, a counter selection step was added (Figure 2A). Briefly, the amplified phage library from the previous round was added to blocked control wells without B7-H3 protein and incubated at room temperature for one hour. Subsequently, unbound phages were taken out and incubated with B7-H3-coated wells for another one hour. The bound phages in the B7-H3-coated wells were then washed, eluted, and designated as the positive selection pool (Figure 2A). Out of 92 randomly selected clones, 12 demonstrated specific binding to B7-H3 in the phage ELISA assay (Figure 2B). The Sanger sequencing of these positive clones revealed a uniform sequence, identifying the predominant binder named A6. To identify typical unspecific binders, bound phages in the control wells were also washed, eluted, and designated as a negative selection pool.

Due to the limited throughput of the single-clone screening, NGS was applied to discover more binders (Figure 2C). The diversified region of the nanobodies eluted from each round was PCR-amplified, and the amplicons were examined by NGS. As expected, clone A6 ranked first in multiple NGS datasets throughout the panning process, including the round 1, round 2, and round 3 positive selection pools. In a phage display, sequences can be enriched but do not necessary need to gain specificity to the target antigen. Enrichment is a complex process that could be influenced by various factors, such as enhanced fitness during phage amplification due to the reduced toxicity of the inserted sequences or binding to non-target antigens (e.g., components found in blocking agents). To assess specificity, NGS datasets were gathered for both the positive and negative selection pools from round 3. The ratio of percentages in the positive versus negative selection pools was calculated for each unique sequence and plotted against the corresponding percentage in the positive pool (Figure 2C). The results unveiled a biphasic distribution regarding specificity in the panning experiment (Figure 2C). The majority of enriched sequences were equally abundant in both the positive and negative pools, indicating their lack of binding specificity to the target antigen.

In contrast, forty-three distinct sequences, including A6, exhibited positive-to-negative ratios exceeding ten. As anticipated, the preeminent specific clone within the NGS dataset was A6, representing a percentage surpassing 20% (Figure 2C). It is worth mentioning that the remaining 42 unique sequences all exhibited percentages below 0.1% in the round 3 positive pool, posing challenges for their detection through a single colony phage-ELISA. Thirty-nine of the identified sequences, including A6, exhibited short CDR3 loops, while the other four sequences displayed CDR3 loops of a medium length. Notably, among the sequences with short CDR3 loops, thirty-six showed striking sequence similarities with A6, differing by only 1–2 amino acids from the A6 sequence. To maximize the likelihood of identifying antibodies binding to different epitopes, we chose to advance two sequences featuring short CDR3 loops, characterized by markedly different sequences from A6, as well as two sequences with medium CDR3 loops, for subsequent validation. A total of five unique sequences, including A6, were selected for the expression and binding assay.

### 2.3. Validation of Selected Binders

To validate their binding, these five nanobodies were reformatted as the human IgG1 Fc-fusion bivalent form (hFc), and their binding was tested using indirect ELISA. Among the five clones, only A6 and a13 showed specific binding to B7-H3 (Figure 3A). In contrast, a22 did not show binding to B7-H3, whereas b1 and b2 showed similar binding to both B7-H3-coated wells and blocked control wells. To further compare the binding strength, a dose-dependent binding experiment was also conducted using indirect ELISA. The half-maximal effective concentration (EC50) of A6-hFc and a13-hFc for recombinant 4IgB7-H3 was 0.33 nM and 1.75 nM, respectively. As a positive control, a human MAb named 8H9 showed approximately a two-fold-stronger binding (EC50 = 0.16 nM) compared to A6-hFc (Figure 3B). In addition, the binding interaction was assessed using biolayer interferometry (BLI). In short, A6-hFc and a13-hFc were biotinylated and immobilized on Octet SA Biosensors. Subsequently, kinetic experiments were conducted using a serial dilution of 4IgB7-H3 in the solution phase (Figure 3C). The results indicated that the affinity of a13 was approximately 75 nM, whereas A6 exhibited an affinity of 31 nM. As a positive control, 8H9 showed an affinity of 10 nM. 8H9 and A6 showed much slower dissociation rates with half lives of 2885 s and 2243 s, respectively, compared to that of a13 (240 s) (Table 1). Additionally, neither A6 nor a13 bound to murine B7-H3 (Appendix A).

### 2.4. Epitope Binning

The locations of epitopes significantly influence the mode of action and efficacy of MAbs. The epitope recognized by MAb 8H9 has been determined to cover the FG loop of B7-H3. To investigate the binding sites of A6 and a13, epitope binning was conducted using BLI in a classical sandwich assay. In this assay, MAb1 with a higher affinity should be immobilized on the biosensor. Then, the MAb1-coupled biosensor is loaded with antigens and finally challenged with the sandwiching MAb (MAb2). An additional response (shift in nm) is observed if MAb2 binds to the antigen, indicating that MAb1 and MAb2 bind to non-overlapping epitopes. On the contrary, if two antibodies bind to the same or partially overlapping epitopes, no or very low response will be observed, respectively. This assay works effectively with monomeric antigens. If antigens are multimeric (like 4Ig), MAb2 can bind to multiple subunits of the antigen, leading to a false binning profile. Therefore, the N-terminal and C-terminal fragments of the human B7-H3 ectodomain (N-2IgB7-H3 and C-2IgB7-H3) were recombinantly expressed, with their bindings to 8H9 and A6 tested. Single-dose experiments revealed that both 2IgB7-H3 bound comparably to antibodies (Appendix A). Therefore, C-2IgB7-H3 was randomly chosen for the epitope binning experiment.

In brief, 8H9-hFc was biotinylated and immobilized on the Octet SA Biosensors. The biosensors were then loaded with C-2IgB7-H3 and further challenged with A6-hFc or a13-hFc. The results showed that neither A6 nor a13 competed with 8H9, as evidenced by their interaction with 8H9-bound antigens (Figure 4A and Appendix A). These results suggest that both A6 and a13 target distinct epitopes from 8H9. As a negative control, 8H9 as MAb2 did not induce any changes. In addition, when biotinylated, A6-hFc was immobilized on the sensor, and a13 failed to elicit a response, suggesting that a13 and A6 bind to overlapping epitopes (Figure 4B and Appendix A). Since A6 were bound to human but not murine B7-H3, a sequence alignment was performed among murine B7-H3, human N-2IgB7-H3, and C-2IgB7-H3 to investigate potential binding sites. The alignment revealed a high similarity between human and murine B7-H3 protein sequences, with only 12 different amino acids (Figure 4C). Two of these residues (R127, G130) are located at the 8H9 epitope (Figure 4D). The other ten residues are potential components within the epitope of A6 and a13, which will be examined in our later study.

### 2.5. Characterization of A6

As A6 and a13 were bound to an overlapping site, and A6 showed stronger binding to B7-H3 compared to a13, further experiments were conducted with this clone. The specificity of A6 to B7-H3 was confirmed using Western blot analysis with the HCT116 and Jurkat cell lines (Figure 5A). HCT116 is a human colorectal cancer cell line known to overexpress B7-H3. As expected, Western blot analysis confirmed the specific binding of A6 to B7-H3 in the HCT116 lysate, with 8H9 as positive control (Figure 5A). In contrast, A6 did not bind proteins in the cell lysates of the human leukemia T cell line, Jurkat, which does not express B7-H3 according to the Human Protein Atlas. Flow cytometric experiments using HCT116 cells further confirmed that A6 was bound to B7-H3 in its membrane-anchored form (Figure 5B). To further validate that the binding of A6 to HCT116 was mediated by B7-H3, we created B7-H3 knockdown HCT116 cell pools using the clustered regularly interspaced short palindromic repeats (CRISPR) method. The reduced expression of B7-H3 was confirmed by Western blot at the protein level (Appendix A). Flow cytometric analysis revealed that A6 showed decreased binding to the B7-H3 knockdown HCT116 cell pools compared to HCT116 WT (Figure 5B). No binding was observed for a human IgG1 isotype control to HCT116. The binding of A6-hFc to HCT116 cells is dose-dependent, with an EC50 of approximately 85 nM (Figure 5C).

### 2.6. Antibody-Dependent Cellular Cytotoxicity

Antibody-dependent cellular cytotoxicity has been identified as a vital mechanism employed by IgG1 antibodies to effectively eliminate tumor cells, underscoring its significance in the context of therapeutic antibody efficacy.

To evaluate the ADCC activity of A6, we fused it both to the human IgG1 Fc domain as A6-hFc-WT, and the Fc variant (S239D/I332E) as A6-hFc-SDIE. Both forms of A6-hFc were tested in the ADCC luciferase reporter assay. Jurkat/NFAT-luc/CD16 V158 (InvivoGen, San Diego, CA, USA) cells were used as the effector cells against cancer cells with B7-H3 expression. This is a genetically engineered Jurkat cell line with the human FcγRIIIa (CD16A) V158 allotype and a luciferase reporter under the control of the nuclear factor of activated T cells (NFAT) response elements. Of note, Jurkat cells naturally express a functional NFAT transcription factor, which is involved in the early signaling events of ADCC. A simultaneous binding of the MAbs to the target and this Jurkat reporter cell line leads to the activation of NFAT, which is then monitored by luciferase activity.

As anticipated, both A6-hFc-WT and A6-hFc-SDIE treatment demonstrated a dose-dependent luciferase signal in the ADCC detection system (Figure 6). As a negative control, the IgG1 isotype did not yield any noticeable signal. Notably, A6-hFc-SDIE (EC50, 0.67 nM) exhibited an approximately 100-fold stronger ADCC activity compared to A6-hFc-WT (EC50, 81.5 nM).

## 3. Discussion

Various reasons support the utilization of nanobodies in monoclonal antibody, bispecific, or chimeric antigen receptor (CAR) T cell therapy developments, such as their distinctive single-chain structure, compact size, and ease of expression and production [29]. In this study, we successfully isolated nanobodies from our synthetic phage display library, which used degenerate codon for CDR randomization. One previous study also used degenerate codon for nanobody CDR randomization but in a ribosome display library with a higher library size ($10^{11}$) [7]. Ribosome display is not limited by cell transformation and culture constraints, allowing for a much higher diversity for display, but it remains underutilized compared to phage display or yeast display systems, possibly due to its sub-optimal efficiency and fidelity.

In our workflow of in vitro selection, the phage display library was counter-screened on blocked control wells to reduce the number of unspecific binders. The NGS of both the negative and positive pools enabled the calculation of the specificity scores to explore sequences with low abundance. It is noted that the lack of specificity for the majority of enriched sequences (Figure 2C) is not uncommon, probably because the library was synthetic instead of immune, as observed in previous studies [30,31,32]. In addition, the result showed two of the five selected sequences specifically bound to B7-H3. We also found that 36 clones identified by NGS demonstrated sequence similarities with A6, differing by only one to two amino acids. Similarly, a recent study reported that enriched clones with a high specificity were close homologs, sharing >80% identity in CDR L3 and H3 sequences [30]. The success rate of binder identification based on the NGS specificity score was relatively low. This could be because the selected nanobody sequence adopted different conformations when recombinantly expressed compared to those in the pIII protein fusion form in phage surfaces. Sequencing errors cannot be excluded as one of the potential possibilities of mis-identification.

Recent studies have discovered that the location of epitopes have a significant impact on the mode of action and efficacy of monoclonal antibody [25] and bispecific antibody [33]. Therefore, the development of various antibodies proves invaluable in pinpointing the potent epitope. B7-H3 possesses two distinct domains, namely, IgC and IgV. Although the epitope for most B7-H3 antibodies remains undescribed, two of the well-established clones, 8H9 and 376.96, were identified as binding to the FG loop in the IgV domain [27,28]. This loop has been previously associated with a possible role in inhibiting T cell proliferation [34]. Li et al. (2023) recently generated dromedary camel nanobodies targeting the IgC domain of B7-H3 and demonstrated that CAR-T cells based on one of the nanobody sequence had potent anti-tumor activity against large tumors in a mouse model [28]. Notably, a recent study also reported that murine-derived MAb 7C4 targeted an epitope in the IgC domain involving Q179, resulting in a human B7-H3-specific binding profile similar to A6 [33]. In this study, we found that both A6 and a13 were bound to a distinct region compared to 8H9. However, the exact epitopes of these two nanobodies still await further study. Moreover, the potency of using these nanobodies in different formats of immune therapy, including MAbs, bispecific antibody, and CAR-T cell, will enable the understanding of the significance of targeting this epitope.

Anti-B7-H3 agents, including MAbs, bispecific antibodies, antibody-drug conjugates (ADCs), CAR-T cells, and radioimmunotherapy agents, have exhibited encouraging anti-tumor activity in preclinical models and have recently entered clinical trials for multiple cancer types [11]. B7-H3-specific ADCs have been actively developed to treat solid tumor. For instance, pyrrolobenzodiazepine-conjugated B7-H3 ADCs killed both cancer cells and tumor vasculature, eradicating large established tumors and metastases in a preclinical study [16]. DS-7300a (Daiichi Sankyo, Tokio, Japan), an ADC consisting of a B7-H3-specific MAb conjugated to four topoisomerase I inhibitor particles, is currently being tested in a Phase I/II trial [35]. Additionally, MGC018 (humanized B7-H3 MAb with a cleavable linker-duocarmycin payload, MacroGenics, Rockville, MD, USA) entered a clinical trial based on encouraging data in a preclinical model [36], although it has been terminated due to business reasons.

Another promising B7-H3-targeting immunotherapeutic strategy is represented by Fc-enhanced MAbs. Enoblituzumab (MGA271, MacroGenics, Rockville, MD, USA), a fully humanized MAb bearing an Fc domain engineered to enhance its anti-tumor function, was found in Phase I trials to be well-tolerated [37]. Enoblituzumab is currently being evaluated in a Phase II trial against localized prostate cancer as neoadjuvant [38]. The rationale for using an engineered Fc domain to improve the effector cell function has been supported by the correlation of clinical outcomes of MAb therapies with the natural polymorphisms of Fcγ receptors (FcγRs) in patients. In this study, A6-hFc showed a dose-dependent ADCC effect on B7-H3-expressing cells in the luciferase-based assay. It has been shown that the two point mutants (S239D/I332E) of the Fc domain used in this study enhanced effector-mediated anti-tumor functions via increased affinity for the activating receptor CD16A [39]. Outstandingly, the A6 fused to this enhancing Fc variant further improved the ADCC activity by approximately 100 folds (Figure 6). It is important to note that the current ADCC data were derived from in vitro reporter assays, and further investigation in in vivo models can validate the efficacy of A6 in these IgG formats.

Multispecific antibodies utilizing B7-H3-specific MAbs represent an additional B7-H3 targeting modality. For instance, anti-B7-H3 IgG has been fused to either an anti-CD3 or anti-4-1BB single-chain variable frament (scFv) to construct bispecific antibodies to recruit and activate T cells against cancer cells [33,40]. In another study, an anti-PD-L1 IgG fused to an anti-B7-H3 nanobody was shown to induce a synergetic effect [41]. Anti-B7-H3 antibodies have also been linked to CD16-specific antibodies, which can lead to NK-cell-mediated cancer cell lysis [42,43]. Lastly, B7-H3-targeting CAR-T cells have been generated, and some of them are being evaluated in trials targeting different solid tumors [28,44]. The A6 clone is currently used as the B7-H3-specific arm to develop bispecific and trispecific antibodies in the lab. The synthetic library developed in this study can be further explored to discover nanobodies against different targets in order to ease the development of these multi-specific agents. Murine B7-H3-blocking MAbs were tested in mice, and the result showed that tumor growth reduced with CD8^+^ T and NK-cells’ infiltration density increased [45]. However, the limited information on the B7-H3 receptors hinders the translation of these findings to the clinical setting. Nanobodies discovered in this study can be tested for their ability to suppress human T-cell activation and validated as blocking antibodies to inhibit tumor growths in this context.

In summary, a large synthetic nanobody library was constructed in this study, with the desired CDR positions fully randomized using degenerate codons. This phage display library was successfully applied to isolate B7-H3-specific nanobodies, which were shown to target a distinct region compared to known MAbs. One of the clones, when fused to an enhancing Fc domain, showed a remarkable ADCC efficacy. This demonstrates that the synthetic libraries produced binders with functional activity. Future work will focus on the detailed epitope mapping of these newly isolated nanobodies and characterize their activity using both in vitro and in vivo models. The newly generated synthetic library is especially suited for nanobody discovery tasks where binder generation via immune libraries fails due to self-tolerance, toxic antigens, or insufficient stability of the antigen.

## 4. Materials and Methods

### 4.1. Construction of Synthetic Library

The DNA library of nanobodies was assembled by one-step assembly PCR, similar to a previous design [4]. A set of ten primers, P1, P2, to P10 (Appendix A), were mixed to prepare “mix 9a”, “mix 9b”, and “mix 9c” containing each primer in an equimolar ratio, which only differed in the P9 primers (P9a, P9b, and P9c), to introduce CDR3 regions of variable lengths with 7, 11, or 15 continuous NNB codons. A concentration of each primer at 0.25 µM in the mixed pool produced optimal yields for the assembly PCR with a PrimeSTAR Master Mix (Takara Bio, Dalian, China). PCR products with the correct size were purified by DNA agarose gel extraction and used as templates for attaching a 3×Myc-tag by an overlapping PCR. The final products were purified and mixed in a 1:2:1 molar ratio of short/medium/long CDR3 loops. The mixture was digested with a PstI/BstEII restriction enzyme (NEB, Ipswich, MA, USA) and ligated with linearized pMES4 plasmids (GenBank GQ907248) using T4 DNA ligase (Takara Bio, Dalian, China). The ligation product was subsequently purified and transformed to an electro-competent *E. coli* TG1 strain. Library diversity was estimated by plating serial dilutions of the transformed bacteria.

The transformed TG1 cells were plated on large agar plates. All colonies were scraped off the plates with a 5 mL 2 × Yeast Extract Tryptone (2×YT) medium. After complete resuspension, these TG1 cells were diluted in a 2×YT medium to a initial optical density at 600 nm (OD600) of 0.1 and grew until the OD600 reached 0.6–0.8. To amplify phages, the culture was mixed with helper phage M13KO7 (molar ratio of TG1:M13KO7 = 1:20) and incubated with 0.2 mM Isopropyl β-D-Thiogalactoside (IPTG) and 50 μg/mL Kanamycin with constant shaking overnight at 30 °C. The phage was harvested by precipitation with 20% polyethylene glycol (PEG) 8000/2.5 M NaCl. The precipitated phages were resuspended in a PBS buffer (137 mM NaCl, 2.7 mM KCl, 10 mM Na_2_HPO_4_ and 1.8 mM KH_2_PO_4_, pH 7.4), aliquoted and stored at −80 °C for later use.

Unproductive sequences due to insertion, deletion, or frame shift can be reduced by one round of anti-Myc selection. Protein G magnetic beads (Invitrogen, Carlsbad, CA, USA) were coated with an anti-Myc antibody (clone 9E10) and washed three times with PBS containing PBST. The phage solution ($10^{12}$ CFU) was pre-cleared with the Protein G magnetic beads. The supernatant was then transferred to 34 µL antibody-coated beads and incubated at room temperature for 30 min. The beads were washed three times with PBST and eluted with 50 µL 50 mM glycine (pH 2.5). Eluates were neutralized with 50 µL 1 M Tris-HCl (pH 7.5).

### 4.2. Next-Generation Sequencing for Nanobody Library

#### 4.2.1. Amplicon Preparation for Sequencing

DNA from *E. coli* library was isolated using the plasmid extraction method. Then, $10^{7}$ CFU of *E. coli* was centrifuged at 10,000 rpm for 5 min and then extracted by a Plasmid Mini Kit (Omega Bio-Tek, Norcross, GA, USA). DNA from the phage library was isolated using the sodium iodide–ethanol precipitation method. The steps below were for 500 µL of solutions containing $10^{12}$–$10^{13}$ CFU/mL of phages. Firstly, the phage solution was mixed with a 200 µL PEG/NaCl solution and incubated on ice for 2 h. The solution was then centrifuged at 12,000 rpm for 15 min at 4 °C, and the supernatant was discarded. The pellet was thoroughly dissolved in a 5 M sodium iodide solution (63 µL), 100% ethanol (156 µL) was subsequently added, and the mixture was further incubated on ice for 2 h to precipitate DNA. A second centrifugation (12,000 rpm, 4 °C, 15 min) was conducted to harvest DNA as a white or translucent pellet. Finally, the pellet was resuspended in 70% ethanol (200 µL) to remove residual salts. The solution was centrifuged at 12,000 rpm at 4 °C for 15 min, and the ethanol supernatant was discarded. The pellet was dried for 20 min at room temperature.

The extracted DNA was used for PCR amplification with primers flanking the variable region (10 µL PCR reaction containing 40 ng DNA template, 15 cycles) (Appendix A). The PCR product was examined in a 1.5% (*w*/*v*) agarose gel with Tris-acetate-EDTA buffer. The band corresponding to the expected product was extracted from the gel and purified using an E.Z.N.A.^®^Gel Extraction Kit (Omega Bio-Tek, Norcross, GA, USA). Purified amplicons were sent for sequencing library preparation and NGS in GENEWIZ company (Suzhou, China). NGS was performed with Novaseq paired-end 2 × 250 bp (Genewiz, Suzhou, China). At least one million reads were collected for each sample.

#### 4.2.2. NGS Data Analysis

Paired-end reads were merged to generate complete sequences according to the overlap regions between Read1 and Read2 using FLASH [46]. The adaptor and primer sequences were trimmed using Cutadapt [47]. The Seqkit tool set was used to translate and analyze the sequences [48]. Sequences were filtered to generate intact sequences by estimated amplicon lengths, correct amino acids at the end of the sequencing region, and absence of early stop codon. The percentage of intact sequences was calculated as the number of intact sequences divided by the total number of successfully merged and trimmed sequences. Python scripts, which were in-house written and will be made available upon request, were used to calculate (1) the percentage of each amino acid type in a specific position, (2) the percentage of each amino acid type averaged over all fully randomized sites in the nanobody library, and (3) numbers of each unique sequence in the nanobody library. The figures were created using python’s matplotlib [49].

### 4.3. Expression and Purification of B7-H3

Human 4IgB7-H3 and murine 2IgB7-H3 ectodomain were subcloned from a gene vector (Sino Biological, Beijing, China) into the pcDNA3.1(+) expression vector. The expression vectors were chemically transformed into DH5a *E. coli*. For expressing human 4IgB7-H3 ectodomain, 200 μg endotoxin-free expression plasmids were transfected to 200 mL Expi293F cells (Thermo Fisher, Walthem, MA, USA) with a Polyethylenimine MAX (PEI MAX) transfection reagent (Polysciences, Warrington, PA, USA). Supernatants from transfected cells were collected after 96 h and then filtered through a 0.22 µm Polyethersulfone (PES) filter (NEST, Wuxi, China). The filtered supernatant was mixed with equal volumes of a binding buffer (20 mM NaH_2_PO_4_, 20 mM Na_2_HPO_4_, 500 mM NaCl, 10 mM imidazole, pH 7.4) and purified using HisTrap HP column (Cytiva, MA, USA) in an AKTA Pure protein purification system. Subsequently, the column was washed with the binding buffer until a stable baseline was attained and eluted with an elution buffer (20 mM NaH_2_PO_4_, 20 mM Na_2_HPO_4_), 500 mM NaCl, 500 mM imidazole, pH 7.4). The eluate was concentrated using a Vivaspin centrifugal concentrator (Sartorius, Göttingen, Germany) with a molecule weight cut-off (MWCO) of 30 killo-Dalton (kDa) for human B7-H3 and 10-kDa for murine B7-H3. After the proteins were dialyzed to a PBS buffer, a final polishing of the proteins was conducted using size exclusion chromatography (Superdex 200 10/300 GL). The soluble B7-H3 protein was mixed with a final concentration of 50% glycerol and stored at −80 °C. To analyze the binding epitope, N-terminal 2Ig (residue 29–246) and C-terminal 2Ig (residue 247–458) of human B7-H3 ectodomain were subcloned from a gene vector (Sino Biological, Beijing, China) into the pcDNA3.1(+) expression vector. The expression and purification process was similar to the process described above.

### 4.4. Phage Display Panning

Recombinant 4IgB7-H3 proteins were coated onto a 96-well plate (Thermo Fisher, Walthem, MA, USA, 439454) for phage display panning. Briefly, five wells were each coated with 250 ng of recombinant proteins in PBS overnight at 4 °C. All wells were then washed with PBST three times and blocked with a blocking buffer (PBS with 5% skimmed milk) at room temperature for 1 h. Subsequently, $10^{12}$–$10^{13}$ CFU of blocked phages were added to each well and allowed to bind for 1 h at room temperature. After incubation, the wells were washed 15 times with PBST and 5 times with PBS to clear unbound phages. Finally, 100 µL of 100 mM glycine (pH 2.5) was added to each well and incubated at room temperature for 15 min to elute bound phages. The eluate was immediately neutralized by a 50 µL 1 M Tris-HCl buffer (pH 7.5). A small fraction of the neutralized eluate was serially diluted with a 2×YT medium and used for titer measurements. To produce phages for further panning, the TG1 cells (OD = 0.6–0.8) were infected by the eluate with constant shaking for 60 min and were spread in large agar plates with 100 μg/mL ampicillin for overnight incubation at 30 °C. Phages were packaged from TG1 cells and purified as described above. In the third round, the amplified phages from the second round were first incubated with skimmed-milk-blocked control wells. Unbound phages were then transferred to the B7-H3 protein-coated wells for panning. The washing, elution, and amplification steps were similar to the previous panning steps.

### 4.5. Phage ELISA

After three rounds of selection, recovered TG1 cells were plated, and colonies were randomly picked to prepare single-clone phages for ELISA. In brief, individual colonies were picked and diluted to a 200 µL 2×YT medium with ampicillin in a 96-cell culture plate for overnight growth at 37 °C. The overnight culture in each well was diluted 100 times to a fresh medium and continued incubation at 37 °C. M13KO7 was added when OD600 reached 0.6–0.8. After incubating at 37 °C for 30 min, phage production was induced by the addition of IPTG (0.2 mM) and further incubation at 30 °C for 16 h. In ELISA, each well in the Maxisorp plate (Thermo Fisher, Walthem, MA, USA) was coated with recombinant proteins or a skimmed milk control (250 ng/well) in a PBS buffer. After being blocked with 3% bovine serum albumin (BSA) (Sigma, St. Louis, MO, USA) in PBS, 80 µL of phages were transferred to each well and incubated at room temperature for 2 h. The plates were washed with PBST five times and then incubated with Horseradish peroxidase (HRP) conjugated Anti-M13 antibody (Sino Biological, Beijing, China, 11973) at room temperature for 1 h. Subsequently, the plates were washed with PBST five times and developed by incubating with a tetramethylbenzidine (TMB) substrate (GBCBIO, Guangzhou, China) (50 µL/well) for 3 min. The reaction was quenched by adding 50 µL of 1 M HCl. Absorbance at 450 nm was measured with Multiskan FC microcoder (Thermo Fisher, Walthem, MA, USA).

### 4.6. Expression and Purification of Nanobody-hFc

The nanobody A6 and a13 sequences were subcloned to pFUSE-hIgG1-Fc2 (InvivoGen, San Diego, CA, USA), which was modified to insert a 6xHis-tag at the C-terminal of the Fc domain, to produce a nanobody–human IgG1 Fc fusion antibody (Nanobody-hFc). An Fc-enhanced A6 antibody was constructed to include two mutations in the Fc domain, specifically serine (S) at residue 239 to aspartic acid (D), and isoleucine (I) at position 332 to glutamic acid (E). This mutant was named A6-hFc-SDIE. The nanobody-hFc was produced by an Expi293F transient expression system (Thermo Fisher, Walthem, MA, USA) as described above for recombinant B7-H3. Samples were purified using TALON resin (Takara Bio, Dalian, China) and polished by size exclusion chromatography (Superdex 200 10/300 GL). Purified antibodies were concentrated and buffer exchanged to PBS.

### 4.7. Nanobody-hFc ELISA

Nanobody-hFcs were tested for binding in ELISA against recombinant human 4IgB7-H3. Briefly, protein and skimmed milk control were coated on the Maxisorp plate as described above. On the next day, the plate was blocked with 3% BSA in PBS. The antibodies were diluted with a blocking buffer to 50 nM and added to wells to incubate for 1 h at room temperature. The plate was washed five times with PBST and then incubated for 1 h with an HRP-conjugated goat anti-human IgG-Fc secondary antibody (Proteintech, Chicago, IL, USA, SA00001) diluted 1:5000 in the blocking buffer. After five washes with PBST, the plate was developed as described above for the phage ELISA. In the dose-dependent ELISA experiment, three-fold serial dilutions of antibodies (starting from 300 nM) were added to the plate in duplicates for each concentration of one antibody.

### 4.8. Western Blot

The preparation of HCT116 or Jurkat cell (ATCC) lysate for sodium dodecyl-sulfate polyacrylamide gel electrophoresis (SDS-PAGE): $4\times 10^{6}$ cells were collected with the medium removed and diluted in a 200 µL lysis buffer (150 mM NaCl, 50 mM Tris, 1% TritonX-100, 1% phenylmethylsulfonyl fluorid, pH 8.0). After incubation for 30 min, the sample was centrifuged at 12,000 rpm for 10 min to collect the supernatant. The samples were separated on a 10% SDS-PAGE and transferred to Polyvinylidene fluoride (PVDF) membranes. The membranes were blocked with 5% skimmed milk diluted in PBST overnight at 4 °C and then incubated with A6-hFc (200 nM) for 1 h at room temperature. The membranes were then washed five times with PBST and incubated with an HRP-conjugated goat anti-human IgG-Fc secondary antibody (Proteintech, Chicago, IL, USA, SA00001) diluted 1:5000 in the blocking buffer. For loading control, we used a rabbit monoclonal anti-GAPDH antibody (HuaBio, Hangzhou, China, HA721136) at 1:10,000 dilution as the primary antibody and an HRP-conjugated mouse anti-rabbit IgG (Santa Cruz, Dallas, TX, USA, 1:5000) as the secondary antibody. Detection was performed using Omni-ECL femtolight Substrate (EpiZyme, Shanghai, China, SQ202).

### 4.9. Generation of B7-H3 Knockdown HCT116 Cells

A Cas9 sgRNA plasmid system was used to generate B7-H3 knockdown HCT116 cell pools. A B7-H3 sgRNA target sequence was designed by the gRNA design tool (genscript.com, accessed on 1 June 2023). An sgRNA sequence 5’-TTGATGTGCACAGCGTCCTG-3’ was chosen and cloned into pSpCas9(BB)-2A-Puro (PX459) V2.0 (Tsingke Bio, Beijing, China). Endotoxin-free plasmid (2 μg) was used to transfect 2 mL of HCT116 cells in a six-well cell culture plate with a 6 μg PEI MAX transfection reagent (Polysciences, Warrington, PA, USA). After 48 h, the supernatant was aspirated out, and a fresh culture medium with 2 μg/mL puromycin (BioFroxx, Einhausen, Germany) was added. The cells were incubated for three days under antibiotic pressure. Cells after this antibiotic-resistance screening were transferred to a new plate for expansion and protein expression detection. To detect protein expression, $4\times 10^{6}$ cells were collected to prepare whole cell lysates as described above and quantified by a BCA Protein Assay Kit (CWBio, Beijing, China). Lysates containing 30 μg of proteins were run on 10% SDS-PAGE and transferred to a PVDF membrane. The membrane was blocked with 5% skimmed milk in a TBST buffer (20 mM Tris, pH 7.5, 150 mM NaCl, 0.1% Tween-20) overnight at 4 °C and then incubated with a rabbit polyclonal anti-B7-H3 antibody (Sino Biological, 201526) at 1:1000 dilution, or rabbit monoclonal anti-GAPDH antibody (HuaBio, Hangzhou, China, HA721136) at a 1:10,000 dilution for 1 h at room temperature. An HRP-conjugated mouse anti-rabbit IgG (Santa Cruz, 1:5000) was used as a secondary antibody. The membranes were washed five times with a TBST buffer for 5 min at room temperature. Lastly, the blot was developed by incubating it with an Omni-ECL femtolight Substrate (Epizyme, Shanghai, China).

### 4.10. Flow Cytometry

Flow cytometry was used to assess the binding of A6-hFc to HCT116 or B7-H3 knockdown HCT116 cell pools. A human IgG1, kappa isotype (HG1K-500, Sino Biological), served as the negative control. About $10^{6}$ cells were incubated with 100 nM A6-hFc or isotype control in the dark at 4 °C for 30 min. Cells were washed three times with ice-cold PBS buffer (pH 7.4) and then incubated with an FITC F(ab’)2 goat anti-human IgG Fcγ antibody (Biolegend, San Diego, CA, USA, 398006) in the dark at 4 °C for 30 min. Finally, the sample was analyzed with CytoFLEX (Beckman Coulter, Indianapolis, IN, USA).

For dose-dependent analysis, HCT116 cells ($10^{6}$) were incubated with a 5-fold serial dilution (from 10 μM) of A6-hFc in the dark at 4 °C for 30 min. The cells were washed, incubated with the secondary antibody, and analyzed as described above. The binding activity was represented using normalized mean fluorescence intensity (MFI). The MFIs of each sample were normalized by subtracting the MFI of the negative control and dividing by the difference between the MFI of the antibody at 10 µM and the MFI of the negative control.

### 4.11. Binding Kinetics with Biolayer Interferometer

Antibodies were diluted in a PBS buffer to a final concentration of 5 μg/μL. The antibodies were then incubated with EZ-Link Sulfo-NHS Biotin (Thermo Fisher, Walthem, MA, USA) in a molar ratio of 20:1 (biotin:antibody) on ice for 2 h. Free biotins were removed by centrifugation using Vivaspin centrifugal concentrator (Sartorius, MWCO 5000). The biotinylated antibodies were buffer-exchanged into PBS at a pH of 7.4.

Binding kinetics were studied using the Octet R8 system (Sartorius, Göttingen, Germany). Bindings were performed at 16 °C with shaking at 1000 rpm in a black 96-well plate (Greiner Bio-One, Kremsmünster, Austria) containing 200 μL of solution in each well. A PBST buffer (pH 7.4) containing 10 mg/mL BSA was used for the analyte dilution and washing. Streptavidin (SA) biosensors were rehydrated, equilibrated, and then loaded with biotinylated antibodies (50 nM) for 300 s to reach a shift at around 1.5 nm. The sensors were washed, associated with a three-fold serial dilution of antigens for 300 s, and allowed for antigen dissociation in running buffer for 600 s. Regeneration was repeated three times, with a short incubation (5 s) in glycine (pH 2.5) and neutralization in running buffer (10 s). An empty biosensor was used as a reference. The data were analyzed using Octet BLI Analysis 12.2 software. The sensorgrams were subtracted from the reference well, and the kinetics were analyzed using association and dissociation steps and fitted into a standard 1:1 binding model.

### 4.12. Epitope Binning

The SA sensors were equilibrated to reach baseline and then loaded with biotinylated antibodies (MAb1) at 50 nM, followed by a 180 s wash. Subsequently, the sensors were bound by antigens at 600 nM for 300 s and allowed for dissociation for 600 s. Finally, the sensors were allowed for the binding of the unbiotinylated second antibody (MAb2, 50 nM) for 300 s. The unbiotinylated form of MAb1 was used as a blocking control at the MAb2 stage to demonstrate complete blocking with a fully overlapping epitope.

### 4.13. ADCC Luciferase Reporter Assay

ADCC luciferase reporter assays were performed using Jurkat-Lucia NFAT-CD16 reporter cells (InvivoGen, San Diego, CA, USA), which are effector cells that stably express the FcγRIII receptor and an NFAT-response element-driving expression of luciferase. The ADCC assay buffer contained an RPMI 1640 medium (Thermo Fisher, Walthem, MA, USA) with a 10% fetal calf serum (Nobimpex, Herbolzheim, Germany), supplemented with penicillin and streptomycin (Biological Industries, Göttingen, Germany). HCT116 cells were diluted in the ADCC assay buffer and added to the cell culture plate for 20,000 cells per well, incubated for 24 h at 37 °C with 5% CO_2_. On the next day, supernatants in the well were removed and replaced with 90 µL of the fresh assay buffer. Five-fold serial dilutions (starting from 10 μM) of antibodies were added to each well (20 µL/well) and incubated for 1 h at 37 °C with 5% CO_2_. A human IgG1 kappa isotype (HG1K-500, Sino Biological) served as the negative control. After that, Jurkat-Lucia NFAT-CD16 reporter cells were added (90 µL/well) to a 20:1 effector/target cell ratio. The plate was incubated for 24 h and then centrifuged at 4500 rpm for 10 min. Supernatants (20 μL/well) were transferred to a 96-well white plate and then mixed with 50 μL of QUANTI-Luc™ 4 Reagent (InvivoGen, San Diego, CA, USA). The signal was measured immediately with the LUMINESCENCE program of the GloMax DISCOVER instrument (Promega, WI, USA). The data were fitted to a nonlinear model (three parameters) using GraphPad Prism 8 software.

### 4.14. Sequence and Structural Analysis

Sequence alignment was conducted using ESPript 3.0 [50]. The predicted structure of the N-terminal 2Ig of human B7-H3 was extracted from an AlphaFold protein structure database [51]. The B7-H3 protein expression patterns in different cell lines were extracted from the Human Protein Atlas [52].

## Figures and Tables

**Figure 1 bioengineering-11-00381-f001:**
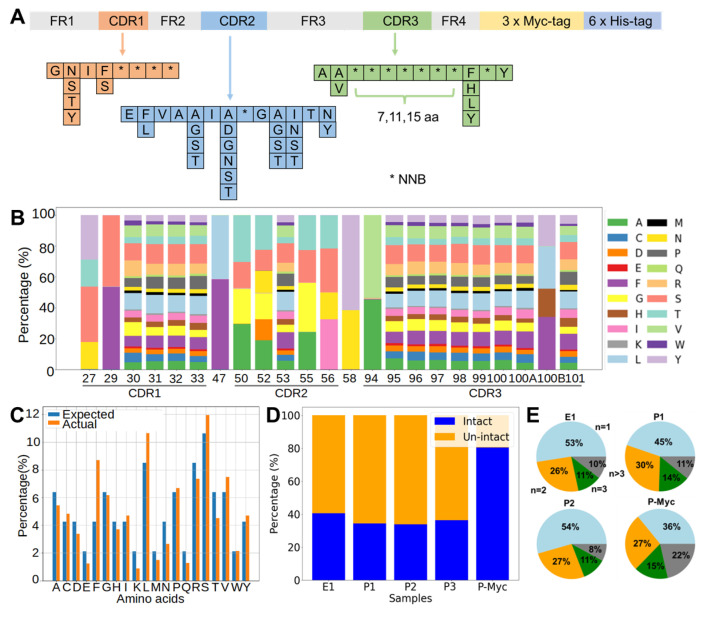
Quality evaluation of synthetic nanobody library by next-generation sequencing. (**A**) Diagram depicting the synthetic nanobody gene. The framework regions, highlighted in gray, remained constant in sequence. Conversely, sections of the CDR loops were subject to variation, with CDR1, CDR2, and CDR3 depicted in orange, blue, and green, respectively. Partial randomization was accomplished using mixed nucleotides, while highly variable regions were randomized using degenerate codons NNB, denoted by asterisks (*). (**B**) For demonstration, amino acid distributions in desired positions were shown for sequences with the short CDR3 in the library. The IMGT definition was used to annotate the CDRs using the AbYsis tool (http://www.abysis.org/abysis/index.html, accessed on 6 June 2023). (**C**) The observed and expected ratios for 20 amino acid types over fully randomized sites in nanobodies with the short CDR3 loop. (**D**) Percentage of intact sequences in NGS data for the *E. coli* library (E1), three batches of independently prepared phage display libraries without anti-Myc tag enrichment (P1, P2, and P3), and the phage library after anti-Myc tag enrichment (P-Myc). (**E**) Pie charts show percentages of sequences with one, two, three, and more appearances in corresponding libraries as (**D**).

**Figure 2 bioengineering-11-00381-f002:**
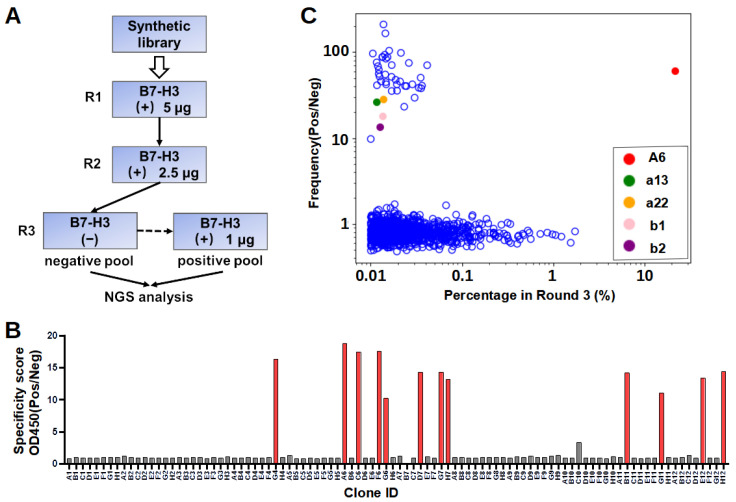
Phage display panning process and NGS mining. (**A**) The panning process is shown schematically with the amount of recombinant proteins used in each round listed. B7-H3-coated wells are denoted by a “+” sign, and skimmed-milk blocked-control wells are denoted by a “−” sign. The solid lines with arrows indicate that the eluted phages from the previous round were amplified prior to the subsequent round. The dashed line with an arrow indicates that unbound phages from the preceding incubation step were transferred directly to the succeeding wells. In round 3, phages eluted from B7-H3 (−) (negative pool) and B7-H3 (+) (positive pool) were amplified and used for NGS analysis. (**B**) Single-colony phage ELISA identified potential specific clones. Phages produced from randomly picked colonies were tested for the binding to 4IgB7-H3 or skimmed-milk-blocked control wells. The specificity score was calculated by dividing the absorbance in an antigen-coated well by the absorbance in the blocked control well. Clones with absorbance below 0.1 in the control wells and a specificity score higher than five were determined to be specific binders. Gray and red bars indicated non-specific and specific binders, respectively. (**C**) NGS result of round 3. The ratio of percentages in the positive versus the negative pools is graphed for each unique sequence against its corresponding percentage in the positive pool. In this analysis, only sequences exceeding 0.01% in the positive pool were considered. Each blue open circle or colored solid circle represented a unique sequence.

**Figure 3 bioengineering-11-00381-f003:**
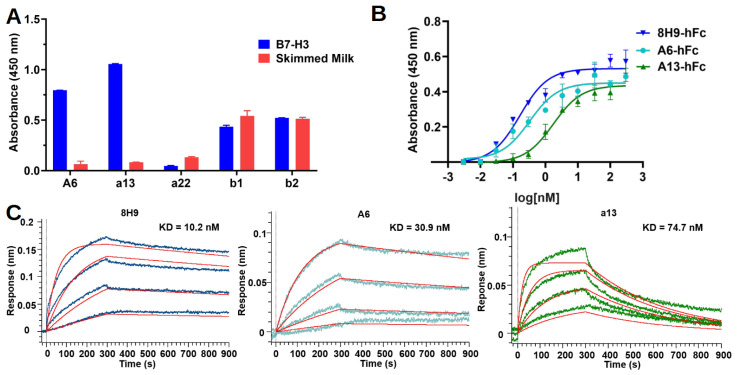
Binding validation for selected clones. (**A**) ELISA showed binding of selected clones in hFc format to 4IgB7-H3-coated wells and blocked control wells. (**B**) A6-hFc and a13-hFc showed dose-dependent binding to recombinant B7-H3 in indirect ELISA assay. B7-H3-specific MAb 8H9 was used as the positive control. (**C**) BLI sensorgrams showed A6, a13, and 8H9 bound to a serial dilution (900 nM, 300 nM, 100 nM, and 33 nM) of recombinant B7-H3. Raw data are in blue, cyan, and green, respectively, and the fitting curves are in red.

**Figure 4 bioengineering-11-00381-f004:**
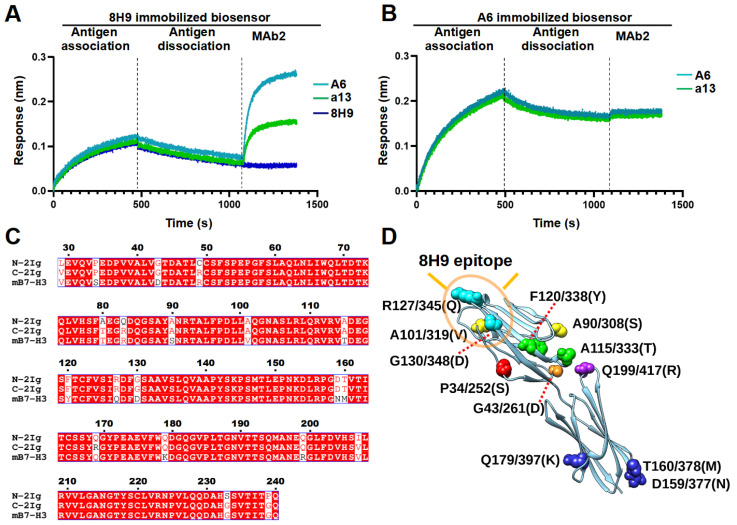
Epitope binning analysis. Streptavidin-coated biosensors were coupled with biotinylated (**A**) 8H9-hFc or (**B**) A6-hFc, followed by antigen C-2IgB7-H3 and incubated with the sandwiching MAb (MAb2). The labels indicate the MAb2 identity. (**C**) Sequence alignment of human and murine B7-H3 with residue numbers provided (referring to the N-terminus of human B7-H3). Residues 1-28 are suggested to be the signal sequence and cleaved during protein maturation. (**D**) Predicted structure of the N-terminal 2Ig of human B7-H3 was extracted from AlphaFold protein structure database, with non-conserved residues between human and murine B7-H3 depicted in spheres of different colors. Residue numbers (in both N- and C-terminal 2Ig) and the corresponding amino acids in human B7-H3 are labeled. The corresponding amino acids in murine B7-H3 are in parentheses.

**Figure 5 bioengineering-11-00381-f005:**
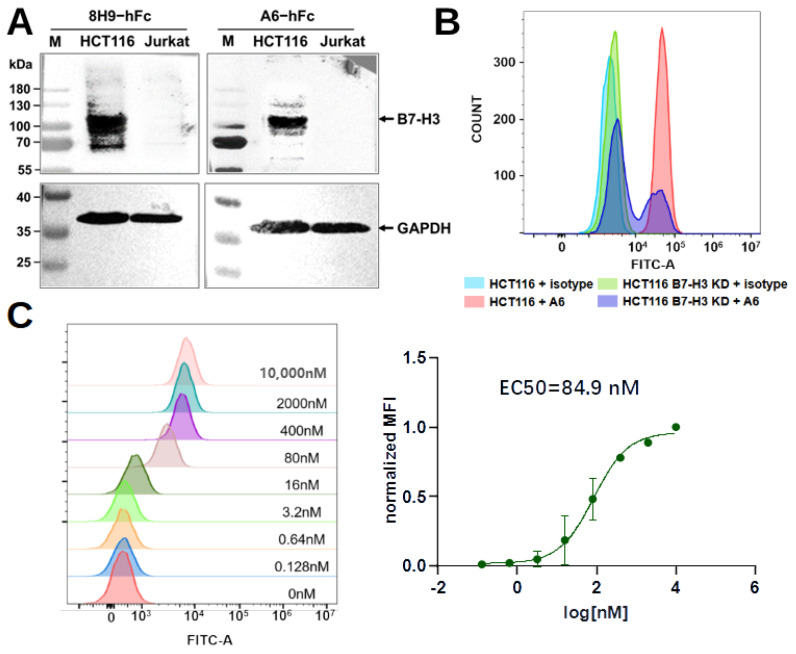
Characterization of A6. (**A**) Western blot analysis of A6 binding to HCT116 and Jurkat cell lysates, using 8H9 as the positive control. Glyceraldehyde-3-phosphate dehydrogenase (GAPDH) was used as a loading control. (**B**) Flow cytometric analysis of A6 binding to HCT116 WT and B7-H3 knockdown (B7-H3 KD) cell pools. A human IgG1 isotype was used as a negative control. (**C**) Left, one representative flow cytometric experiment is shown for the dose-dependent binding of A6-hFc to HCT116. Right, data from three independent experiments were averaged and fitted with a three parameter non-linear model.

**Figure 6 bioengineering-11-00381-f006:**
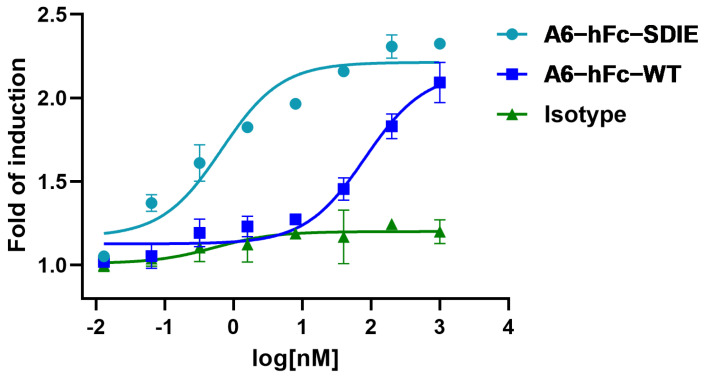
ADCC activity of A6. HCT116 was used as target cells and Jurkat-Lucia NFAT-CD16 (InvivoGen) as reporter cells. The dose-dependent experiment was conducted by measuring the induced luciferase activity of the assay supernatant after 24 h. An IgG1 isotype served as the negative control. The data represent means ± standard deviation (SD) from three independent experiments.

**Table 1 bioengineering-11-00381-t001:** Kinetic rate constants obtained by the analysis of biolayer interferometry data.

Antibody	$k_{a}$ (1/Ms)	$k_{d}$ (1/s)	$K_{D}$ (nM)	$K_{D}$ Error (nM)	Half Life (s)
8H9	2.36 × 10^−4^	2.40 × 10^−4^	10.16	0.29	2885
A6	1.00 × 10^−4^	3.09 × 10^−4^	30.90	0.63	2243
a13	3.87 × 10^4^	2.89 × 10^−3^	74.70	0.96	240

## Data Availability

The data and source code generated during and/or analyzed during the current study are available from the corresponding author on reasonable request.

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
