# Peer review of "Development of a Human B7-H3-Specific Antibody with Activity against Colorectal Cancer Cells through a Synthetic Nanobody Library"

_bioengineering, 2024, doi:10.3390/bioengineering11040381_

Round 1

Reviewer 1 Report

Comments and Suggestions for Authors

On manuscript on Li et al on application on synthetic biology techniques on antibody technology. Although manuscript carries relevant findings on this area on lacks some clarifications. On began on the main advantages on application on nanobodies on comparison on monoclonal antibodies? I mean on main reason on specificity? On about the costs and benefits? Another remark refers on a novel application on synthetic biology techniques on cancer treatment? Also, on main limitations on application on nanobodies on cancer treatment? On main specificities on treatment?  

Finally, on limitations on binding on nanobodies on comparison on common monoclonal antibodies on any health problems?

Reviewer 2 Report

Comments and Suggestions for Authors

The authors developed a synthetic nanobody library targeting human B7-H3. One of these nanobodies, A6, demonstrated potent antibody-dependent cell-mediated cytotoxicity against colorectal cancer cells.

The manuscript is well structured and contains very explicative figures with good quality. I believe that the manuscript contains a sufficient level of data and quality to be published in a bioengineering journal.

The results section contains a large number of cited references. Please consider that when the authors chose to present their results separate from the discussion, the results section should not include references, aiming for a complete separation of the results from the discussion.

Author Response

Thank you for pointing this out. We have now moved those references from the Result section to other relevant parts.

In line 88, we have moved “, similar to a previous design [4]” to line 353 (Materials and Methods) “The DNA library of nanobodies was assembled by one-step assembly PCR, similar to a previous design [4].”

In line 146, “This phenomenon is not uncommon, as observed in previous studies [29-31], probably due to that the library was synthetic and not immune.” We have now moved it to the Discussion part. In line 268-270, we wrote “It is noted that the lack of specificity for the majority of enriched sequences (Figure 2C) is not uncommon, probably due to that the library was synthetic instead of immune, as observed in previous study [29–31].”

In line 169-171, “As a positive control, a human MAb named 8H9 showed approximately two-fold stronger binding (EC50 = 0.16 nM) compared to A6-hFc (Figure 3B) [27]”. As 8H9 has been mentioned in the Introduction section and referenced, therefore, the citation here is deleted.

In line 181-182, “The epitope recognized by MAb 8H9 has been determined to cover the FG loop of B7-H3 [27].” Similarly, 8H9 has been mentioned in the Introduction section and referenced, therefore, this citation here is deleted.

In the legend of Figure 4, “(C) Sequence alignment [32] of human ….”. The citation [32] described the method used to align the sequence. Here we moved the sequence alignment method to the Materials and Methods section. In line 589, we wrote “Sequence alignment was conducted using ESPript 3.0 [32]”.

In the legend of Figure 4, “(D) Predicted structure of the N-terminal 2Ig of human B7-H3 was extracted from AlphaFold protein structure database [33],….”. The citation [33] described the source of the predicted structure of the N-terminal 2Ig of human B7-H3. Here we moved this information to the Materials and Methods section. In line 589-590, we wrote “The predicted structure of the N-terminal 2Ig of human B7-H3 was extracted from AlphaFold protein structure database [33]”.

In line 218, “A6 did not bind proteins in the cell lysates of the human leukemia T cell line, Jurkat, which does not express B7-H3 according to the Human Protein Atlas (proteinatlas.org) [34].” This citation 34 was moved to the Materials and Methods section. We wrote “B7-H3 protein expression pattern in different cell lines were extracted from the Human Protein Atlas (proteinatlas.org) [34]”.

In line 231-235, “Fc mutants that can enhance ADCC activity have been discovered, and one of the two point mutants, S239D/I332E, was shown to enhance effector-mediated anti-tumor function via increased affinity for the activating receptor CD16A [35]. A previous study showed that Rituximab incorporating this substitution resulted in a significantly lower EC50 for B cell depletion in cynomolgus monkeys [35].” These sentences introduced the source of the enhancing Fc variant that we used in the study. We now moved this part to the Discussion. In line 316-320, we wrote “It has been shown that the two point mutants (S239D/I332E) of the Fc domain used in the study enhanced effector-mediated anti-tumor function via increased affinity for the activating receptor CD16A [35]. Outstandingly, A6 fused to this enhancing Fc variant further improves the ADCC activity by approximately 100 folds (Figure 6).”

In line 246-248, “Efficacy given by this type of assays have been well correlated with the standard ADCC assay using human peripheral blood mononuclear cell (PBMC) [36]”. We removed this sentence since it has been addressed in the Discussion.

Reviewer 3 Report

Comments and Suggestions for Authors

With the emergence of new generation Ab technologies like single chain, nanobody (Nb), bispecific, Fc engineered, biosimilar, mimetic, and conjugated Ab, the healthcare and pharmaceutical industry has been ingenious in developing highly specific mAb treatments for various diseases like cancer, autoimmune and infectious diseases. Synthetic Nb libraries, however, have emerged as an attractive alternative to animal immunization to select antigen-specific Nb.  Using a combination of next-generation sequencing and single-clone validation, the authors identified two Nbs that specifically bind B7-H3. In addition, one of the clones demonstrated potent antibody-dependent cell-mediated cytotoxicity (ADCC) against a colon cancer cell line. The work has been thorough and in compliance with all the necessary validation criteria. All in all, the article could be recommended for publication as is.

Author Response

We would like to express our sincere gratitude for the valuable feedback you have provided.

Round 2

Reviewer 1 Report

Comments and Suggestions for Authors

Authors on adressed comments and now manuscript warrants publication on Bioengeneering.